# Diabetes and Cancer: Risk, Challenges, Management and Outcomes

**DOI:** 10.3390/cancers13225735

**Published:** 2021-11-16

**Authors:** Rabia K. Shahid, Shahid Ahmed, Duc Le, Sunil Yadav

**Affiliations:** 1Department of Medicine, University of Saskatchewan, Saskatoon, SK S7N 5A2, Canada; rabia.shahid@saskhealthauthority.ca; 2Saskatoon Cancer Center, Saskatchewan Cancer Agency, University of Saskatchewan, Saskatoon, SK S7N 4H4, Canada; duc.le@saskcancer.ca (D.L.); sunil.yadav@saskcancer.ca (S.Y.)

**Keywords:** diabetes mellitus, diabetes, cancer, hyperglycemia, targeted therapy, cancer therapy

## Abstract

**Simple Summary:**

Diabetes mellitus is a common disease in patients with cancer. It is a risk factor for certain cancers such as pancreatic, liver, colon, breast, and endometrial cancer. Furthermore, several new cancer treatments or the use of steroids may unmask underlying diabetes or aggravate preexisting diabetes. Evidence suggests that patients with cancer and diabetes have higher cancer-related mortality. Moreover, concurrent complications associated with diabetes in patients with cancer may influence the choice of cancer therapy. This review highlights the relationship between diabetes and cancer and various aspects of the management of diabetes in patients with cancer.

**Abstract:**

Background: Diabetes mellitus and cancer are commonly coexisting illnesses, and the global incidence and prevalence of both are rising. Cancer patients with diabetes face unique challenges. This review highlights the relationship between diabetes and cancer and various aspects of the management of diabetes in cancer patients. Methods: A literature search using keywords in PubMed was performed. Studies that were published in English prior to July 2021 were assessed and an overview of epidemiology, cancer risk, outcomes, treatment-related hyperglycemia and management of diabetes in cancer patients is provided. Results: Overall, 8–18% of cancer patients have diabetes as a comorbid medical condition. Diabetes is a risk factor for certain solid malignancies, such as pancreatic, liver, colon, breast, and endometrial cancer. Several novel targeted compounds and immunotherapies can cause hyperglycemia. Nevertheless, most patients undergoing cancer therapy can be managed with an appropriate glucose lowering agent without the need for discontinuation of cancer treatment. Evidence suggests that cancer patients with diabetes have higher cancer-related mortality; therefore, a multidisciplinary approach is important in the management of patients with diabetes and cancer for a better outcome. Conclusions: Future studies are required to better understand the underlying mechanism between the risk of cancer and diabetes. Furthermore, high-quality prospective studies evaluating management of diabetes in cancer patients using innovative tools are needed. A patient-centered approach is important in cancer patients with diabetes to avoid adverse outcomes.

## 1. Introduction

Diabetes is a common disease among cancer patients, yet few prospective studies have evaluated its impact on cancer care. Approximately 10% of the North American population have diabetes, of which about 95% have type 2 diabetes [1,2]. Diabetes is a risk factor for certain solid malignancies such as pancreatic, liver, colon, breast, and endometrial cancer [3,4]. Cancer patients with diabetes face unique challenges. Several novel cancer therapies or the concurrent use of steroids may unmask underlying diabetes or aggravate preexisting diabetes [5,6,7]. Evidence suggests that cancer patients with diabetes have higher cancer-related mortality and patients with diabetes are at an increased risk of various infections and infection-related morbidity and mortality [4,8,9,10]. In general, individuals with diabetes have an annual mortality rate twice that of the general population of similar age [11,12,13]. Cancer patients with diabetes may receive suboptimal care that could lead to inferior outcomes [14]. Moreover, concurrent complications associated with diabetes such as chronic renal insufficiency, cardiovascular disease, peripheral neuropathy, and chronic infection in diabetic patients may influence the choice of cancer therapy. Therefore, these factors may limit the use of certain drugs or optimal dosing that may lead to lower benefits and shorter survival. In the context of the current scarcity of literature on diabetes and cancer, this review has been undertaken to provide an overview of the epidemiology, cancer risk, outcomes, cancer treatment-related hyperglycemia and the management of diabetes in cancer patients.

## 2. Literature Search

A literature search using PubMed was performed and the studies that were published in English involving human subjects prior to July 2021 were assessed. The following keywords were used in combination: ‘diabetes mellitus’, ‘diabetes’, ‘hyperglycemia’, ‘diabetic ketoacidosis’, ‘prediabetes’, ‘insulin’, ‘oral hypoglycemic agent’, and ‘cancer’, ‘malignancy’, ‘targeted therapy’, ‘immunotherapy’, or ‘cancer therapy’.

## 3. Classification

Type 1 diabetes is characterized by a lack of insulin production by the pancreas mostly due to the autoimmune-mediated destruction of pancreatic beta cell. In contrast, type 2 diabetes is characterized by combination of peripheral insulin resistance and dysfunctional insulin secretion by pancreatic beta cells, which results in high but inappropriately low circulating insulin concentrations for the level of glycemia [15]. Most people with diabetes have type 2 diabetes and only about 5-10% of individuals have type 1 diabetes. Both type 1 and type 2 diabetes are multifactorial diseases involving several genetic and environmental factors. However, obesity and other life style factors play an important role in the development of type 2 diabetes [15]. Diabetes can also develop secondarily to medication or from an underlying illness. Various cancer directed agents that are discussed below could unmask underlying diabetes or cause secondary diabetes [16,17,18]. Among secondary diabetes, type 3c diabetes mellitus is secondary to diseases of the exocrine pancreas including pancreatitis and pancreatic cancer [19].

## 4. Epidemiology

Cancer and diabetes have several common risk factors, such as obesity, smoking, aging, physical inactivity and unhealthy diet [20]. The global incidence and prevalence of both cancer and diabetes are rising. It is estimated that in 2017 more than 450 million people were living with diabetes worldwide, whereas the current incidence of cancer excluding non-melanoma skin cancer is about 19.3 million [21,22]. Cancer is primarily a disease of older adults, with the median age of a cancer diagnosis being 66 years [23]. Approximately 60% of patients that are newly-diagnosed with cancer are aged ≥65 and about 25% of persons aged ≥65 develop diabetes mellitus [24]. Overall, 8–18% of cancer patients have diabetes as a comorbid medical condition, with the percentage depending on the cancer site. A case of new-onset diabetes could be marker of underlying pancreatic cancer. A population-based study found that 1% of subjects aged ≥50 years with new-onset diabetes had underlying pancreatic cancer [25].

Given the high prevalence of diabetes in the general population, screening for diabetes is important in cancer patients prior to commencement of systemic anticancer therapy. In addition, patients with diabetes should be undergo age-appropriate cancer screenings [26].

## 5. Link between Diabetes and Cancer

### 5.1. Mechanism of Cancer Risk in Diabetes

Hyperglycemia, insulin resistance, elevated insulin and insulin-like growth factor-1 (IGF-1) levels, inflammatory cytokines, dyslipidemia, increased leptin, and decreased adiponectin have been attributed to the increased risk of cancer in patients with diabetes [27]. Insulin is a member of a family of growth factors that includes IGF-I and IGF-II [28]. Insulin is most widely known for its metabolic effects, but it also has important mitogenic effects [27,28]. Hyperinsulinemia due to insulin resistance, hyperglycemia, dysregulations of sex hormones, oxidative stress, and inflammatory cytokines potentially play a role in the development of cancer [29,30], and elevated glucose levels lead to the proliferation of various solid tumor cell lines [31,32]. Both the liver and the pancreas are exposed to a high insulin concentration, as insulin is transported to the liver via the portal vein following release from pancreatic β-cells [29]. In addition, underlying obesity associated with diabetes has cancer-promoting effects secondary to increased peripheral estrogens, increased pro-mitogen cytokines and growth factors related to excess fat tissue [27]. Moreover, fat-associated chronic inflammation, the release of inflammatory cytokines and the generation of reactive oxygen species (ROS) result in cell damage, cancer growth, and invasiveness [27,33].

### 5.2. Role of the Microbiome in Cancer and Diabetes

It is worth mentioning the evolving role of the gut microbiome in the development of diabetes and cancer [34,35,36]. The human digestive tract has trillions of microorganisms including bacteria, viruses, yeast, protozoa, and fungi that interact with each other and the host and play a role in homeostasis and the development of diseases. Cancer cells depend on glucose and glycolysis as a key source of energy. The complex interaction between the gut microbiome, sugar compounds, insulin resistance, and cancer is not fully known. Nevertheless, evidence supports that extrinsic factors including diet, the environment, and sugar compounds disrupt the gut microbial diversity and function, a phenomenon known as dysbiosis, and increase the risk of insulin resistance, diabetes and several types of gastrointestinal, hepatobiliary and head and neck cancers such as liver, pancreatic, esophageal, and most notably colorectal cancer [34,35,37,38,39]. The underlying mechanism leading to cancer is not known and may involve chronic inflammation, impaired host immune function, production of carcinogenic byproducts, and disruption of programmed cell death [36,40]. In addition, there is emerging evidence that certain infection such as candidiasis, which is more common patients with diabetes, has been associated with an increased risk of cancer [37,41,42].

### 5.3. Epidemiological Link between Diabetes and Cancer

Most published studies evaluating the association between cancer and diabetes have not made a distinction between the two main types of diabetes. As type 2 diabetes is more common than type 1diabetes, the study populations have been largely comprised of individuals with type 2 diabetes. Nevertheless, studies that focused on individuals with type I diabetes and were adequately powered have shown an excess risk of cancer and cancer-related mortality for overall and various site-specific cancers in both type 1 and type 2 diabetes [43,44].

Stocks et al. examined six European cohorts including 274,126 men and 275,818 women, showing that for a 1 mmol/l increment in blood glucose the relative risk of cancer in men is 1.05 (1.01–1.10) and in women is 1.11 (1.05–1.16) [45]. Diabetes and hyperglycemia are associated with an elevated risk of developing pancreatic, liver, colorectal, breast, bladder and endometrial cancer, and a lower risk of prostate cancer [3,4,46,47]. A meta-analysis of 151 cohorts including over 32 million people, 1.1 million cancer cases, and 150,000 cancer deaths showed a strong association between type 2 diabetes and liver, pancreatic, and endometrial cancer incidence, and pancreatic cancer mortality [3]. A population-based case control study using the Surveillance Epidemiology and End-Results Program (SEER) data showed that diabetes was an independent risk factor for hepatocellular cancer, with an odds ratio of 3.08 (95% CI: 2.74–3.46) [48]. Likewise, a meta-analysis of 40 studies showed that women with diabetes have a relative risk of development of breast cancer of 1.27 (95% CI: 1.16–1.39) [49]. Another meta-analysis of 35 cohort studies showed that diabetes was associated with an independent relative risk of pancreatic cancer of 1.94 (95% CI: 1.66–2.27) [50]. However, the association between pancreatic cancer and diabetes is complex; a recent diagnosis of diabetes may be an early sign of pancreatic cancer and may reflect insulin resistance that is elicited by pancreatic cancer [51]. A comprehensive umbrella review of 27 meta-analyses of type 2 diabetes and various cancers demonstrated that diabetes is associated with a 10% relative risk of cancer (RR: 1.10; 95% CI: 1.04–1.17) [52].

### 5.4. Diabetic Pharmacotherapy and Risk of Cancer

Insulin is the mainstay of treatment for type 1 diabetes, but for type 2 diabetes, both hypoglycemic medication and insulin are used. There have been concerns regarding whether exogenous insulin, especially insulin glargine, could increase the risk of cancer [53,54]. The ORIGIN trial investigators randomized 12,537 subjects with impaired glucose tolerance, or type 2 diabetes to insulin glargine or standard care and examined cardiovascular outcomes and incidence of cancer [55]. The investigators did not find an increased risk of cancer with the use of basal insulin glargine (hazard ratio for cancer, 1.00). At the present time, it is not conclusive whether there is an increased risk of malignancy from exogenous insulin that warrants a change in the clinical practice [56]. With respect to the association between other anti-diabetes medications and cancer incidence and mortality, the current data are conflicting and there is no definite evidence that they either increase or decrease the risk of cancer [57]. For example, retrospective cohort studies have shown that incretin-based therapies involving glucagon-like peptide 1 receptor agonists (GLP-1 RAs) and dipeptidyl peptidase 4 (DPP-4) inhibitors in patients with diabetes have been associated with an increased risk of pancreatic cancer [58]. However, a meta-analysis of 33 studies did not confirm a positive relationship between incretin-based therapies and pancreatic cancer and a causal relationship between incretin-based therapies and pancreatic cancer or pancreatitis is no known [59,60].

On the contrary, preclinical studies have demonstrated antineoplastic effects of metformin, an oral biguanide, and this has subsequently been supported by several large epidemiological studies and meta-analyses [61,62]. However, to date, no randomized clinical trial has confirmed its protective effect in cancer.

## 6. Diabetes Mellitus & Cancer Outcomes

Diabetes may negatively influence prognosis because of other associated comorbidities, as well as the risk of non-cancer-related causes of death [12,14]. Continuous exposure to hyperglycemia and elevated concentrations of circulating insulin might stimulate cancer growth and progression, leading to poorer prognosis. Patients with established diabetes, elevated insulin concentrations, elevated insulin-like growth factors, and abnormal glucose tolerance experience higher mortality and recurrence rates after diagnosis and treatment of underlying cancer [9,63,64,65]. Several cohort studies have suggested a higher cancer-related mortality among patients with diabetes and certain solid malignancies [8,9]. Epidemiological studies have shown that cancer mortality rises in linear fashion with increasing glucose concentrations [65,66]. A systemic review of 23 studies demonstrated that patients diagnosed with cancer who have pre-existing diabetes are at a 41% increased risk for long-term, all-cause mortality compared with those without diabetes [63]. Another meta-analysis of 15 studies showed that underlying diabetes was associated with increased odds of postoperative mortality across all cancer types (OR = 1.85 [95% CI: 1.40–2.45]) [67]. Goodwin et al. evaluated insulin levels in 512 nondiabetic women with early-stage breast cancer [64]. The highest levels of fasting insulin were significantly associated with distant recurrence and death, even after adjustment for body mass index (BMI), age, hormonal receptor status, and other known prognostic factors in breast cancer. A recent pooled analysis of 3 adjuvant randomized trials involving 5922 patients with stage II and III colon cancer showed that diabetes was associated with 21% increased risk of cancer relapse and 29% increased risk of mortality [68].

The National Health and Nutrition Examination Survey (NHANES II) Mortality Study found that people with diabetes had an elevated risk of dying of infections [69]. In addition, there is evidence that patients with hyperglycemia have a significantly higher in-hospital mortality rate than subjects with normoglycemia [70]. In an adjuvant colon cancer trial, patients with diabetes experienced an elevated incidence of diarrhea, and were more likely to have leukopenia, thrombocytopenia, elevated creatinine, and hypocalcemia [71]. There is also evidence that patients with cancer and underlying diabetes may receive suboptimal care that could compromise their outcomes. For example, a Dutch study showed that diabetic patients with esophageal, colon, breast, and ovarian cancer received less aggressive cancer treatment than those without diabetes [14].

## 7. Anti-Cancer Therapy & Hyperglycemia

Over the past two decades, treatment landscapes of most solid and hematological malignancies have changed, with the emergence of targeted and immunotherapy. In contrast to conventional cytotoxic agents that have minimal or no direct effect on hyperglycemia, several novel targeted compounds and immunotherapy agents can cause hyperglycemia (Table 1). The rate of hyperglycemia varies with underlying cancer type, comorbid illnesses, concurrent medications, targeted agent use and treatment schedule. Immunotherapies such as PD/PD-L and CTLA-4 inhibitors can reduce the production of insulin from islet cells as in type 1 diabetes mellitus, whereas targeted treatments such as PI3K/mTOR inhibitors can induce insulin resistance similar to that seen in type 2 diabetes mellitus [6,72].

### 7.1. Immunotherapy

Immune checkpoint inhibitors including PD/PDL-1 and CTLA-4 inhibitors are a major breakthrough in the management of cancer [73]. These drugs activate T-cells and enhance the immune response against malignant cells, but at the same time can cause autoimmune phenomena including autoimmune endocrinopathies such as hypothyroidism, hyperthyroidism, adrenal insufficiency or hypophysitis. Uncommonly, autoimmune diabetes mellitus has also been reported following the use of a combination of CTLA-4 and PD/PD-L1 inhibitors such as ipilimumab and nivolumab, or the PD/PD L1 inhibitors nivolumab or pembrolizumab alone [72].

### 7.2. Targeted Therapy

Targeted therapy aims to affect signaling pathways or proteins that are aberrant in the malignant cells and inhibit cell proliferation, regulate the cell cycle or induce apoptosis [74]. However, various targeted agents have been associated with a high risk of hyperglycemia. Among these agents, PI3K/mTOR inhibitors are the most likely to cause hyperglycemia.

#### 7.2.1. PI3K/mTOR Inhibition

The PI3K/Akt/mTOR pathway is commonly activated in various solid cancers [75]. PI3K/mTOR pathway inhibition also affects insulin signaling, resulting in insulin resistance.

##### Mammalian Target of Rapamycin (mTOR) Inhibitors

Everolimus is an orally administered rapamycin analog and mTOR inhibitor that has been approved for the management of advanced breast cancer, renal cell cancer, and pancreatic neuroendocrine tumors. Everolimus can cause hyperglycemia in 10–50% of patients. However, severe hyperglycemia is not common and was only noted in about 10% cases [7]. A meta-analysis of nine randomized clinical trials involving 3879 patients with various solid tumors showed that everolimus was associated with a high risk of all-grade (RR = 2.60, 95% CI: 2.03–3.31) and high-grade (RR = 3.0, 95% CI: 1.72–5.23) hyperglycemia. The incidences of all- and high-grade everolimus-related hyperglycemia were 6.8% (95% CI: 3.4–13.2%) and 2.5% (95% CI: 1.2–4.9%), respectively [76]. Temsirolimus is a potent, highly-specific inhibitor of mTOR that is administered intramuscularly. It has been used in the management of advanced solid malignancies such as renal cell cancer. The incidence of grade 3 or higher hyperglycemia with temsirolimus is about 9% [77].

##### Phosphatidylinositol 3-Kinase (PI3K) Inhibitors

Alpelisib is a PI3K3CA inhibitor. In the randomized control SOLAR-1 trial, alpelisib has shown efficacy in combination with fulvestrant, an endocrine therapy, in women with advanced hormone receptor-positive breast cancer with a PIK3CA mutation [78]. However, women who received alpelisib had an any-grade hyperglycemia rates of 63.7%, and a grade 3 hyperglycemia rate of 32.7%. In addition, 3.9% of patients who received alpelisib developed grade 4 hyperglycemia compared to 0.3% in the placebo group.

##### AKT Inhibitors

Ipatasertib is a selective inhibitor of AKT, a frequently activated protein kinase in human cancers. In a phase I dose escalation trial in patients with various solid tumors, hyperglycemia was noted in 13% of patients [79]. Capivasertib is another pan-AKT kinase inhibitor that was evaluated in a phase I trial with or without fulvestrant in women with metastatic breast cancer [80]. The monotherapy was associated with all-grade hyperglycemia in 45% patients and grade 3 hyperglycemia in 30% of patients.

#### 7.2.2. Insulin-like Growth Factor Type 1 Receptor (IGF-1R)

IGF-1R is partially homologous to the insulin receptor and activates the Ras/MAPK/ERK and PI3K/AKT/mTOR pathways [81]. It promotes cell proliferation and resistance to apoptosis [82]. IGF-1R inhibitors have been evaluated in various solid tumors [83,84,85]. IGF-1R inhibitors include both monoclonal antibodies and small molecules. Hyperglycemia is found to be common side effect of IGF-1R inhibitors.

##### IGF-1R Monoclonal Antibodies

Dalotuzumab (MK-0646) is a humanized IgG1 monoclonal antibody against IGF-1R that does not cross-react with the insulin receptor [86]. In a randomized trial, addition of dalotuzumab to irinotecan and cetuximab in patients with metastatic colorectal cancer did not improve survival and was associated with high rate of grade 3 hyperglycemia compared to the placebo (21.0% vs. 5.2%) [85].

##### Small Molecule Inhibitors of IGF-IR and the Insulin Receptor

Linsitinib is an oral, small molecule, dual inhibitor of IGF-1R and the insulin receptor that was evaluated in a placebo-controlled phase 3 trial in patients with advanced adrenocortical tumors [87]. Linsitinib did not improve survival compared to the placebo. Overall incidence of grade 3 hyperglycemia with linsitinib was 2%, which was much lower than with the monoclonal antibody.

##### Other IGF-1R Inhibitors

Ganetespib is a highly potent heat shock protein 90 inhibitor that blocks multiple oncogenic pathways, including IGF-1R, EGFR, vascular endothelial growth factor, c-MET, and human epidermal growth factor receptor 2 (HER2). In a phase I trial involving patients with hepatocellular carcinoma, ganetespib-related hyperglycemia along with lipasemia were the most common grade 3/4 adverse effects and were reported in 21% of patients (overall 64% patients experienced any-grade hyperglycemia) [88]. Yet, in a subsequent phase 3 trial in patients with advanced lung cancer, ganetespib-related hyperglycemia was not reported as an adverse effect [89]. Ceritinib is a potent oral tyrosine kinase inhibitor of ALK and a less potent inhibitor of IGF-1R. In a phase I study, 10% of patients with lung cancer who were treated with a 750 mg dose of ceritinib experienced any grade of hyperglycemia [90].

IGF-1R inhibitors are currently not in use in clinical practice and trials are ongoing with monoclonal antibody and small molecule inhibitors to determine their role in the management of refractory cancer.

#### 7.2.3. Epidermal Growth Factor Receptor (EGFR) Inhibitors

The EGFR family includes HER2/neu (ErbB2), HER3 (ErbB3), and HER4 (ErbB4). Tyrosine kinase EGFR inhibitors are available as the monoclonal antibodies cetuximab and panitumumab, or small molecules such as geftinib, erlotinib, afatninib and osimertinib, which are commonly used in the management of gastrointestinal and EFGR mutant lung cancers, respectively [91,92]. In general, these agents do not cause hyperglycemia, as the EFGR pathway is not directly involved in glucose metabolism. Rociletinib is a third generation EGFR inhibitor that in a phase I/II trial has demonstrated efficacy in patients with EGFR-mutant non-small cell lung cancer with the T790M resistance mutation who progressed on standard anti-EFGR treatment [93]. Unlike other agents, the rate of grade 3 hyperglycemia in patients who received therapeutic doses of rociletinib was 22%, and it was the most common grade 3 adverse event. The rate of all-grade hyperglycemia was dose dependent (35% of the patients treated with a dose of 500 mg vs. 67% treated with a dose of 1000 mg).

#### 7.2.4. BCR-ABL Multi-Targeted Tyrosine-Kinase Inhibitors

Nilotinib, dasatinib, and ponatinib are next-generation multi-targeted tyrosine-kinase inhibitors that have been used in the management of chronic myelogenous leukemia (CML) and Philadelphia chromosome–positive acute lymphoblastic leukemia and proven to be more effective than imatinib [94]. They are associated with a high risk of hyperglycemia. For example, in a phase 3 trial involving 846 patients with CML, the rates of all grade and grade 3 or 4 hyperglycemia were 36% and 6%, respectively, with 300 mg nilotinib, and 41% and 4%,respectively, with 400 mg nilotinib compared with 20% and 0% respectively, with imatinib [95]. Hyperglycemia is thought to be related to the development of tissue insulin resistance and compensatory hyperinsulinemia [96].

### 7.3. Hormone Therapy

#### 7.3.1. Anti-Estrogen Therapy

Case-control and case-cohort studies have shown that postmenopausal women with breast cancer treated with anti-estrogen therapy have a high incidence of insulin resistance and diabetes [57,97,98]. For example, a case-cohort study involving breast cancer survivors showed that adjuvant tamoxifen was associated with a 2.2 fold risk of diabetes whereas aromatase inhibitors were associated with a 4.2 fold risk of diabetes [98]. 

#### 7.3.2. Androgen Deprivation Therapy

Androgen deprivation therapy with a gonadotropin-releasing hormone analog or antagonist with or without an oral antiandrogen is used commonly in the management of prostate cancer. The use of these agents has been linked increasingly to insulin resistance metabolic syndrome and an increased incidence of diabetes [57,99,100].

#### 7.3.3. Somatostatin Analogs

Octreotide and lanreotide are somatostatin analogs that have shown efficacy in neuroendocrine tumors including secretory insulinomas, VIPomas, glucagonomas and other non-functional neuroendocrine tumors [101,102,103]. All-grade hyperglycemia has been noted in 2% to 27% cases.

### 7.4. Corticosteroids

Corticosteroids are commonly used in cancer patients as a part of treatment regimen for lymphomas, leukemias, myeloma, and solid tumors such as prostate cancers, as adjunctive agents for nausea and vomiting, supportive measures to improve appetite, and to reduce edema in patients with metastasis to the brain and spinal cord [57,104,105,106]. Corticosteroids are known to induce hyperglycemia [107,108]. In general, patients treated with glucocorticoids have an odds ratio for new-onset diabetes mellitus of about 1.5 to 2.5 [5]. Reduced insulin sensitivity with increased glucose production and inhibition of the production and secretion of insulin by pancreatic β-cells are considered to be the key mechanism of glucose intolerance [5]. For cancer patients with known diabetes, the use of corticosteroids requires close monitoring of blood glucose concentrations and appropriate intervention to prevent complications related to hyperglycemia. Patients on oral hypoglycemics may require short-term insulin during the period of therapy. Patients using insulin should adjust dosages based on frequent monitoring of serum glucose concentrations.

### 7.5. Miscellaneous Compounds

Asparaginase is used commonly in the management of acute lymphoblastic leukemia. Hyperglycemia is common with the use of asparaginase and in pediatric patients with cancer, it varies from 2.5% to 23% [109]. Diazoxide is a potassium channel activator and blocks insulin secretion by opening potassium channels of β cells, resulting in hyperglycemia. It has been used in the management of insulinoma [110].

## 8. Factors That Influence the Choice of Cancer Therapy in Patients with Diabetes

In addition to poorly controlled blood glucose, the choice of anticancer therapy in patients with diabetes could be influenced by underlying chronic renal insufficiency, cardiovascular disease, peripheral neuropathy, and chronic infection as described below. These factors limit the use of certain drugs or their dosage, which can lead to lower response rates, inferior outcomes, and shorter survival. Furthermore, the presence of diabetic-related comorbid illnesses also affects the eligibility to participate in clinical trials with novel agents.

### 8.1. Renal Disease

Kidney impairment is a relatively common complication of diabetes [111]. Diabetic patients with kidney impairment are usually not appropriate candidates for nephrotoxic agents. Alternative drugs or reduction in the dosage are required for drugs cleared by the kidney or for drugs metabolized in the liver and later excreted by the kidney. For example, patients with diabetes and germ cell tumors who have kidney impairment will require adjustment in cisplatin dosage [112]. Likewise, underlying kidney impairment increases the risk of toxicity related to capecitabine, an antimetabolite that is commonly used in gastrointestinal cancers [113].

### 8.2. Cardiovascular Disease

A wide range of chemotherapy agents have been associated with cardiotoxicity, including arrhythmias, myocardial necrosis causing a dilated cardiomyopathy, and vasospasm or vasoocclusion resulting in angina or myocardial infarction [114]. Furthermore, targeted therapy, endocrine therapy and radiation increase the risk of cardiovascular complications. Many patients with diabetes have pre-existing heart disease or are at high risk of development of cardiac toxicity that can influence the choice of appropriate treatment in such patients [111]. Adjuvant trastuzumab in women with HER2-positive breast cancer has been associated with significant reduction in recurrent breast cancer and mortality, however it is contraindicated in individuals with left ventricle dysfunction as it increases the risk of symptomatic cardiac dysfunction [115].

### 8.3. Neuropathy

Peripheral neuropathy is a common complication of diabetes and may affect more than 50% of patients [116]. Peripheral neuropathies are the dose-limiting side effects of several chemotherapeutic agents, such as platinum compounds (cisplatin, carboplatin, oxaliplatin), vinca alkaloids (vincristine, vinblastin, vinorelbine), taxanes (docetaxel, paclitaxel), bortezomib and thalidomide [117]. The presence of diabetic neuropathy limits the use of such agents. Adjuvant use of oxaliplatin-based therapy in patients with stage 3 colon cancer has been associated with a reduced rate of recurrence and a better overall survival [118]. Hence, omission of oxaliplatin in diabetic patients with node-positive colon cancer and neuropathy could increase their risk of recurrent cancer.

## 9. Screening & Diagnosis of Diabetes in Cancer Patients

Up to a third of people with diabetes are undiagnosed. The stress of the illness and treatment may unmask underlying diabetes in patients with cancer. The diagnosis of diabetes is per criteria established by the American Association of Diabetes and the World Health Organization [119]: In an asymptomatic individual, diabetes is diagnosed if they have any of the following criteria: fasting blood glucose ≥7.0 mmol/L (126 mg/dL) or a 2-hour blood glucose level of ≥11.1 mmol/L (200 mg/dL) following a 75 g oral glucose tolerance test or a hemoglobin A1C (HbAIC) value ≥ 6.5% (48 mmol/mol). In symptomatic individuals, diabetes is diagnosed with a random blood glucose level of ≥11.1 mmol/L (200 mg/dL).

Screening of patients with cancer with fasting or random blood glucose and HbA1c is important before embarking on systemic therapy. This is especially important in patients who have risk factors for diabetes such as a high body mass index (BMI), physical inactivity, a family history of diabetes, and history of gestational diabetes, or those who will be treated with targeted agents that are associated with hyperglycemia. Patients with elevated fasting glucose concentrations should have formal testing for diabetes and appropriate treatment. Patients with cancer who are started on targeted therapy or steroids that may induce hyperglycemia and diabetes should have education on self-monitoring of glucose and a baseline glucose level. Given the fact that steroid-related hyperglycemia and diabetes are mostly diagnosed during the postprandial state postprandial blood glucose with or without HbA1c should be performed as a screening test in patients taking steroid [120].

## 10. Management of Diabetes in Cancer Patients

### 10.1. Multidisciplinary Management

Few prospective studies are available on the management of cancer patients with diabetes. Multidisciplinary management of diabetes in patients requiring cancer treatment, involving a diabetes specialist, educator, dietician, pharmacist, and psychosocial support professional, in collaboration with the cancer care team, is vital in order to avoid diabetes-related complications and thereby inferior outcomes. Diabetic patients on anticancer therapy require management of hyperglycemia, fluids and electrolytes, along with management of hypertension, cardiovascular complications, infection, and autonomic neuropathy of the gastrointestinal tract. Patient education and self-management training, interdisciplinary team counseling, home blood glucose monitoring, monitoring side effects and patient’s adherence to prescribed treatment, and ongoing education are important components of management of diabetes in cancer patients. Clinicians must consider the cardiac, renal, and neurological complications that are commonly associated with diabetes.

### 10.2. Management of Hyperglycemia

Hyperglycemia in cancer patients exacerbates dehydration that may result from diarrhea and decreased oral intake because of anorexia, nausea, and mucositis, which are common in cancer patients. It can predispose to acidosis and worsen the side effects of cancer therapy [121,122]. Patients with cancer and diabetes require close monitoring of blood glucose during their cancer treatment, and health outcomes could potentially be improved by ensuring optimal blood glucose during treatment [123,124]. Patients on oral hypoglycemic agents may require a short course of insulin during cancer therapy, whereas patients using insulin may need further dose adjustment. Basal insulin inhibits hepatic glucose production overnight and between meals, and prandial or mealtime insulin promotes glucose uptake into muscles following a meal. Treatment-related complications and reduced oral intake, especially in patients with advanced cancer, could increase the risk of hypoglycemia from insulin and oral hypoglycemic agents; hence, less aggressive measures are warranted in such patients.

Diabetic ketoacidosis is a life-threatening complication of diabetes that is associated with hyperglycemia, ketonemia, and acidosis. Some patients with cancer may present with diabetic ketoacidosis as the initial manifestation of underlying diabetes mellitus [125,126]. These patients require rapid correction of hypovolemia and electrolyte abnormalities along with low-dose insulin infusion. In most cases, cancer treatment can be resumed after resolution of ketoacidosis as along as an optimal blood glucose level can be achieved.

Management of targeted treatment-related type 2 diabetes mellitus includes diet, exercise, oral hypoglycemic agents with or without insulin (Table 2). Metformin does not cause hypoglycemia and is the treatment of choice for sustained grade 1, grade 2, and asymptomatic grade 3 hyperglycemia. However, it is not recommended in patients who are at risk of lactic acidosis, such as those with significant impairment of liver or kidney function. Furthermore, metformin can cause gastrointestinal side effects and its use may be limited in patients with significant gastrointestinal side effects of cancer therapy. Metformin 500 mg twice a day or Metformin XR at a dose of 500 mg may be considered for mild increases in fasting plasma glucose, and can be titrated to a maximum dose of 2,000 mg daily over a period of several weeks. The treatment goals include a fasting plasma glucose level of < 8.9 mm/L (160 mg/dL), a random plasma glucose level of < 11.1 mm/L (200 mg/dL), and HbA1c ≤ 8% for the prevention of hyperglycemia-related symptoms and complications including infection, osmotic diuresis, and hypercoagulability, as well as hypoglycemia [123]. If fasting glucose level is higher than 14 mm/l (250 mg/dl), cancer treatment interruption can be considered until blood glucose is better controlled (Figure 1). If hyperglycemia is not optimally controlled at the maximum tolerated dose of metformin, another oral agent such as dipeptidyl-4 inhibitors, glitazones, or a sulfonylurea may be considered before commencement of insulin. Sodium glucose transporter-2 (SGLT-2) inhibitors may also be used, however patients on SGLT-2 inhibitors should be monitored closely for urinary tract infection and candiadiasis. In addition, there are reports of euglycemic diabetic ketoacidosis related to SGLT-2 inhibitors [127]. Glucagon like peptide (GLP)-1 agonists that are available in oral and injection forms are another option, however their gastrointestinal side effects and weight loss can be troublesome in cancer patients. If hyperglycemia is not controlled with glucose lowering agents, insulin is the preferred option [123,124]. Rapid-acting insulin prior to meals facilitates flexible meal times. Healthy diet, physical activity and optimal body weight should be encouraged for cancer patients with diabetes.

For most patients with hyperglycemia related to checkpoint inhibitor treatment, immunotherapy can be continued along with the management of diabetes. However, for patients who develop diabetic ketoacidosis secondary to immunotherapy, it will be an individual decision whether or not to continue immune checkpoint inhibitor treatment based on the severity of adverse effects, disease burden and overall health of the patient. Regardless, immunotherapy should be paused until diabetic ketoacidosis is resolved.

### 10.3. Management of Fluid and Electrolyte Balance

Factors that influence the fluid intake or volume loss in diabetic patients are severe mucositis from chemotherapy or radiation, nausea and vomiting from highly emetogenic chemotherapy such as a platinum compounds, and diarrhea caused by agents such as 5-FU, irinotecan and targeted therapies. Hyperglycemia in diabetic patients is further exacerbated by dehydration resulting from treatment-related side effects. Optimal glycemic control along with periodic intravenous hydration and judicious use of anti-emetics and anti-diarrheal agents can prevent complications from dehydration including diabetic ketoacidosis and hyperosmolar hyperglycemic syndrome [128,129].

### 10.4. Management of Hypertension

Many targeted therapies such as anti-vascular endothelial growth factor (VEGF) agents including bevacizumab, sunitinib, and sorafenib can cause hypertension [130]. Uncontrolled hypertension in a diabetic patient may aggravate or unmask renal insufficiency. Close monitoring and management of blood pressure is therefore important in diabetic patients who are on targeted therapy. Furthermore, surveillance and early intervention of hypertension is especially important in cancer survivors with diabetes to reduce morbidity and mortality and lower the risk of associated target organ damage.

### 10.5. Management of Cardiovascular Complications

Heart disease, especially coronary heart disease, is a major cause of morbidity and mortality in patients with diabetes mellitus [131,132,133,134]. Compared with patients without diabetes, patients with diabetes more often have multi-vessel disease and episodes of silent ischemia [111,131]. Patients with diabetes may also have a higher risk of other complications, including arrhythmias, cardiogenic shock, heart failure, renal failure, and recurrent myocardial infarction [132,133,134]. Patients with diabetes with a history of coronary artery disease and congestive heart failure on anticancer therapy require close follow-up. Dyspnea with no obvious cause in a patient with cancer and complicated diabetes should be considered an angina equivalent, and workup for underlying undiagnosed coronary artery disease including electrocardiogram (ECG), cardiac enzyme tests, and stress tests should be performed. It is imperative to pay special attention to preventive strategies for cancer survivors with diabetes to reduce future risk of cardiovascular complications, including weight management, regular exercise, treatment of hyperlipidemia, and optimal glucose control [134].

### 10.6. Management of Infection

Patients with diabetes are at higher risk of infection from various pathogens involving various organ sites than the general population (Table 3). Soft tissue, skin, and mucosal infections, including superficial fungal infections and urinary tract infections, are common. In addition, patients with diabetes are at high risk of serious infection such as pyomyositis, emphysematous cholecystitis, malignant external otitis, and necrotizing fasciitis. Hyperglycemia is associated with dysfunctional immune responses, impaired circulation, sensory peripheral and autonomic neuropathy, and skin and mucosal colonization with pathogens such as *Staphylococcus aureus* and *Candida* species predispose patients with diabetes to infection [135,136]. Anticancer therapy further increases infection risk and complications by causing bone marrow suppression, poor wound healing, and breaks in the mucosa. There are several factors that predispose patients with diabetes to *Candida* spp. colonization and infection, including higher salivary glucose levels, reduced salivary flow, microvascular degeneration, epithelial cell surface adhesion, and impaired candidacidal activity of neutrophils, all of which are further aggravated by the immunosuppressive effects of cancer treatment [137]. *Candida* spp. infection has also been associated with an increased risk of cancer including oral cancer [138].

Patient and family education, prophylactic use of antibiotics, growth factors, appropriate nutrition and physical activity may reduce the risk of serious infection in diabetic patients [139,140]. A diabetic patient with febrile neutropenia should be treated with a broad-spectrum antibiotic such as cefepime, carbapenem, or piperacillin-tazobactam. The addition of vancomycin can be considered in the presence of hypotension, mucositis, skin or catheter site infection, history of MRSA colonization, recent quinolone prophylaxis and overall clinical deterioration. An antifungal agent should be considered if neutropenia is expected to persist for more than 5 to 7 days and the patient is persistently febrile without an obvious source of infection.

### 10.7. Management of Diabetic Autonomic Neuropathy of the Gastrointestinal Tract

Diabetic autonomic neuropathy may involve the cardiovascular, genitourinary, and the neuroendocrine system as well as the upper and lower gastrointestinal tract [141,142]. Patients with diabetic gastroparesis can experience significant nausea and vomiting during anticancer therapy. Primary treatment of gastroparesis includes dietary changes (low fat diet and frequent small meals) and administration of antiemetic and prokinetic agents. Diarrhea and rarely steatorrhea can occur with diabetic enteropathy and can be exacerbated during cancer treatment. Fluid electrolyte balance, optimizing glycemic control, treating underlying infection, and anti-diarrheal agents are the corner stones of treatment.

## 11. Survivorship Care

As expected, diabetes is more frequent in cancer survivors compared to the general population [143]. Furthermore, cancer survivors with diabetes have a higher rate of comorbid illnesses compared with survivors with no history of diabetes [144]. As mentioned above cancer survivors with diabetes are at high risk of both acute and chronic toxicities from cancer treatment. Chemotherapy and endocrine therapy especially increase the risk of cardiovascular disease including heart disease, hypertension, kidney disease, and stroke, as well as osteoporosis [132,133,134,145]. In addition, cancer survivors with comorbid illnesses including diabetes, experience overall poorer health and a higher rate of disability [143,146]. Therefore, ongoing surveillance and screening for vascular comorbid illnesses related to diabetes and early intervention are key for cancer survivors with diabetes. Furthermore, continuing emphasis and behavioral intervention on life style factors and promotion of regular aerobic physical activity, weight management, a healthy diet, stress management, reducing alcohol consumption, smoking cessation and self-management of diabetes are important to reduce future complications [147].

## 12. Future Directions

Although the association between diabetes and carcinogenesis has been suggested by several studies, the underlying mechanism is not well known. Oxidative stress may be the place where metabolic and oncogenic processes cross paths and may result in oncogenesis [148,149]. Treatment of diabetes could further compound this, as certain antidiabetic drugs may increase the risk of oncogenesis while others may decrease the risk [150]. However, a definite causal relationship between antidiabetic medication and cancer has not yet been established. Multiple factors potentially contribute to the progression of cancer in patients with obesity and type 2 diabetes, including hyperinsulinemia and insulin-like growth factor I, hyperglycemia, dyslipidemia, adipokines and cytokines, and the gut microbiome [151]. A better understanding of the underlying mechanism between diabetes, and cancer risk is needed, along with large prospective, biomarker-driven population-based studies to explore the causal relationship between the duration and severity of diabetes and different cancers. Conclusive evidence may help to implement primary prevention, early detection/cancer screening, and effective therapeutic measures. High-quality prospective studies using innovative tools are needed to determine whether more rigorous management of insulin resistance and hyperglycemia in patients with cancer improves their outcomes. Moreover, establishment of a national or an international database of patients with cancer and diabetes, identification of personal, tumor-related, and contextual factors that are associated with inferior outcomes, and an increased focus on health services and the implementation of research are important for better outcomes.

## 13. Conclusions

Diabetes and hyperglycemia are associated with an elevated risk of developing of many cancers. Hyperglycemic and diabetic patients experience higher mortality and recurrence rates after diagnosis with, and treatment for, cancer. Presence of renal insufficiency, neuropathy, cardiac dysfunction, and chronic infection in diabetic patients with cancer limit the use of certain drugs or their dosage, which can lead to lower response rates and shorter survival. Clinicians must consider the cardiac, renal, and neurologic complications commonly associated with diabetes in patients undergoing anticancer therapy. Patients on targeted therapy can be managed with appropriate glucose lowering agents without the need of discontinuation of cancer treatment in most cases. In addition, promotion of physical activity, healthy diet and optimal body weight are especially important in diabetic cancer survivors. A patient-centered approach by a multidisciplinary team comprised of various health care professionals and development of clinical pathways are vital to avoid diabetes-related complications and thereby inferior outcomes.

## Figures and Tables

**Figure 1 cancers-13-05735-f001:**
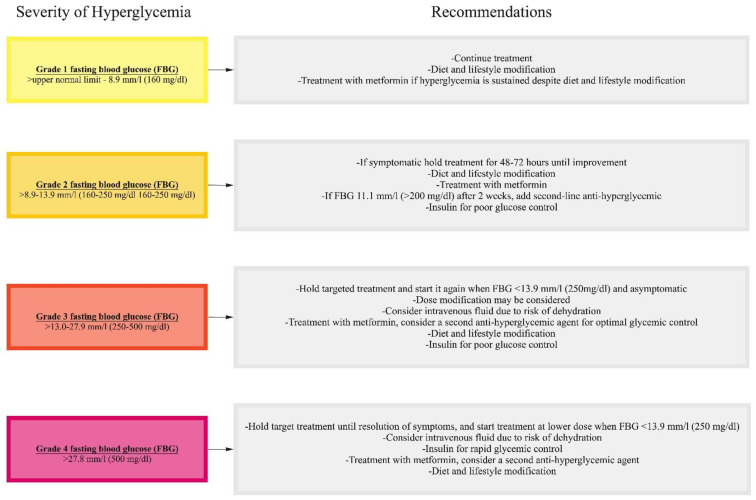
Management of targeted treatment-related hyperglycemia.

**Table 1 cancers-13-05735-t001:** Cancer drugs and their mechanism causing hyperglycemia.

Anticancer Drugs	Mechanism of Action
Targeted Agents	
IGF-1R inhibitor IGF-1R-specific monoclonal antibodies (dalotuzumab) Small-molecule inhibitors of IGF-1R and IR (linsitinib) Other inhibitors of IGF-1R (ganetespib, ceritinib)	Inhibit IGF-1R, which is partially homologous to the IR, and block the activation of the Ras/MAPK/ERK and PI3K/AKT/mTOR pathways and thereby block cancer cell growth and proliferation
PI3K/mTOR inhibition mTOR inhibitors (everolimus, temsirolimus) PI3K3CA inhibitor (alpelisib) AKT inhibitors (ipatasertib, capivasertib)	Inhibit the PI3K/Akt/mTOR pathway and thus interfere with malignant cell growth, but may lead to hyperglycemia by interrupting the intracellular response to insulin, causing decreased glucose transport, decreased glycogen synthesis, and increased glycolysis
BCR-ABL inhibitors (nilotinib, dasatinib, and ponatinib)	Multi-targeted TKIs that inhibit BCR-ABL and other TKIs such as KIT, PDGFR, DDR, and CSF-1R
Anti-EFGR (rociletinib)	Block EGFR pathway and affect downstream signaling cascades, namely the RAS/MAPK/ERK, PI3K/Akt, and JAK/STAT pathways
Immunotherapy	
Anti CTLA-4 (ipilimumab) Immune checkpoint inhibitors (nivolumab, pembrolizumab)	Activate T cells and enhance the immune response against malignant cells; treatment could cause autoimmune phenomena including autoimmune diabetes mellitus
Hormone Therapy	
Somatostatin analogues (octreotide and lanreotide)	Bind predominantly to the somatostatin receptors and suppress insulin secretion
Anti-estrogen therapy (tamoxifen, aromatase inhibitors)	Decrease insulin secretion via inhibition of antiapoptotic effects of estradiol on pancreatic β-cells and increase insulin resistance via elevated triglyceride levels and fatty liver
Anti-androgen therapy (bicalutamide)	Increase insulin resistance by modulating androgen and androgen receptor signaling pathways in the liver and adipose tissue
Miscellaneous	
Asparaginase	Hydrolyzes serum asparagine to nonfunctional aspartic acid and ammonia, depriving tumor cells of asparagine
Diazoxide	Nondiuretic benzothiadiazine that inhibits insulin secretion

BCR-ABL, breakpoint cluster region gene–Abelson proto-oncogene; CTLA, cytotoxic T lymphocyte-associated antigen; EGFR, epidermal growth factor receptor; ERK, extracellular signal-related kinase; IGF-1R, insulin-like growth factor type 1 receptor IR, insulin receptor; MAPK, mitogen-activated protein kinase; PI3K/mTOR, phosphatidylinositol 3-kinase/mechanistic target of rapamycin; TKI, tyrosine kinase inhibitor.

**Table 2 cancers-13-05735-t002:** List of current drugs that are used in the management of diabetes mellitus.

Medication	Mechanism of Action
Biguanides (metformin)	Improve hepatic insulin resistance via decreasing the hepatic glucose output
Second-generation sulfonylureas Glyburide, glipizide, glimepiride	Stimulate endogenous insulin secretion through inhibition of potassium channels in pancreatic cells; most effective in early stages of diabetes when insulin secretion is still working
Meglitinides Repaglinide, nateglinide	Insulin secretagogues that stimulate insulin release by inhibiting potassium channels in the pancreas on a different site from sulfonylureas; work much faster than other secretagogues and can be taken more effectively before meals
Thiazolidinediones Rosiglitazone, pioglitazone	Activate PPARG and improve metabolic control in type 2 diabetes through the improvement of insulin sensitivity in adipose tissue, muscle, and the liver
Glucosidase inhibitors Acarbose, miglitol, voglibose	Inhibit α-glucosidase at the brush border of the small intestine and affect the digestion of complex carbohydrates, resulting in lower postprandial blood glucose
Bromocriptine mesylate	A sympatholytic dopamine D_2_ receptor agonist that exerts inhibitory effects on serotonin turnover in the central nervous system
GLP-1 agonists Exenatide, liraglutide, lixisenatide, dulaglutide, semaglutide	Bind to GLP-1 receptors to restore pancreatic β-cell sensitivity to glucose and to increase β-cell mass
DPP-4 inhibitors (gliptins) Sitagliptin, saxagliptin, linagliptin, vildagliptin, alogliptin	Block GLP-1 degradation
SGLT-2 inhibitors Empagliflozin, canagliflozin, dapagliflozin, ertugliflozin	SGLT2 is expressed in the proximal renal tubules and mediates glucose reabsorption; SGLT2 inhibitors promote the renal excretion of glucose and thereby reduce the serum glucose level
Amylinomimetics Pramlintide	Regulate postprandial spikes in blood glucose by slowing gastric emptying and digestion, promoting satiety, and inhibiting glucagon secretion
Insulin/insulin analogs	Similarly to endogenous insulin, exogenous insulin increases the uptake of glucose into cells, stimulates glycogen synthesis, and inhibits glucagon

DPP-4, dipeptidyl peptidase-4; GLP-1, glucagon-like peptide 1; PPARG, peroxisome proliferator-activated receptor gamma; SGLT-2, sodium glucose transporter-2.

**Table 3 cancers-13-05735-t003:** List of common infections and their etiologies in patients with diabetes.

Organ System	Infection Type	Organisms
Soft Tissues and bones	Diabetic foot, osteomyelitis & septic arthritis	Aerobic Gram-positive cocci including *staphylococcus aureus**,* *streptococcus agalactiae**,* *streptococcus pyogenes**,* gram-negative bacilli, and anaerobic organisms
	Necrotizing fasciitis	*S. aureus*, *S. pyogenes*, *enterobacteriaceae*, *vibrio* species, *aeromonas* species, *salmonella* species, and anaerobic organisms
	Cellulitis	*S. aureus, S. pyogenes, corynebacterium jeikeium, pseudomonas aeruginosa* *(ecthyma gangrenosum)*
Genito-urinary tract	Urinary tract infection: cystitis, urethritis, pyelonephritis,	*Escherichia coli*, *klebsiella* species and other *enterobacteria*, *acinetobacter* species, *P. aeruginosa*. *S. agalactiae*, *candida* *albicans*, and other yeasts
	vulvovaginitis	*Candida albicans*, and other yeasts, *gardnerella vaginalis*, *mycoplasma hominis*
Respiratory tract	Pneumonia	*S. pneumoniae*, *S. aureus*, *K. pneumoniae* and other Gram-negative, bacilli, *legionella* species, influenza virus, mycobacterium tuberculosis complex
Head and neck	Mucormycosis (zygomycosis)	*Rhizopus* species, *mucor* species
	Malignant otitis externa	*P. aeruginosa*, aspergillus species, and other fungi
	Endophthalmitis	*E. coli*, *K. pneumoniae*
	Periodontal disease	Oral commensals organisms, *porphyromonas gingivalis*, *tannerella* *forsythia*,
Gastrointestinal	Cholecystitis	*Enterobacteriaceae*: *E. coli*, other species, anaerobic organisms, *Candida* species
	Typhlitis	*Clostridium* species, enterobacteriacae, *bacteroides fragilis*, *candida* species
	Perianal abscess	Polymicrobial (Gram-positive cocci, gram-negative bacilli and anaerobes)
Bacteremia and sepsis	Community-acquired and hospital-acquired	*viridans Streptococci*, *S. aureus*, *S. pneumoniae*, enterobacteriacae, *E. coli*, klebsiella species, *pseudomonas* *aeruginosa*, *candida albican*, *enterococci*, and others

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
