# Peer review of "Diabetes and Cancer: Risk, Challenges, Management and Outcomes"

_cancers, 2021, doi:10.3390/cancers13225735_

Round 1
Reviewer 1 Report
This is an extensive review text on the topic of cancer and diabetes. This MS is well organized with adequate references. My only topic of concern deals with insulin use and incretin drug use. Some years ago, there were concerns on the risk for cancer among insulin Glargine users. This concern has been met, especially with the ORIGIN study (NEJM, 2014). Nonetheless, risk for cancer is a matter of interest for all new insulin analogs. Also, the incretin (DPP IV inhibitors and GLP1 receptor agonists) anti-diabetic drugs were investigated because of theoretical risk for pancreatic cancer due to their effects on exocrine pancreatic ductal cells. These drugs are not recommended for patients with diabetes and antecedent pancreatic diseases.
Author Response
Reviewer 1
Comments
This is an extensive review text on the topic of cancer and diabetes. This MS is well organized with adequate references. My only topic of concern deals with insulin use and incretin drug use. Some years ago, there were concerns on the risk for cancer among insulin Glargine users. This concern has been met, especially with the ORIGIN study (NEJM, 2014). Nonetheless, risk for cancer is a matter of interest for all new insulin analogs. Also, the incretin (DPP IV inhibitors and GLP1 receptor agonists) anti-diabetic drugs were investigated because of theoretical risk for pancreatic cancer due to their effects on exocrine pancreatic ductal cells. These drugs are not recommended for patients with diabetes and antecedent pancreatic diseases.
Thanks very much for overall positive comments.
- We appreciate very much the suggestion and have elaborated the section on “Diabetic pharmacotherapy and risk of cancer” in the revised manuscript. We have reviewed additional clinical data about relationship between insulin glargine and cancer, provided a reference to the Origin study (reference 53) and discussed the data about risk of pancreatic cancer with incretin.

Reviewer 2 Report
Dear Authors:
In the manuscript by Shahid et al. the authors summarize recent data and research achievements on the relation and mechanisms between diabetes and breast cancer risk. I have just a few suggestions:
1. Some citations are missing:
In page 11, line 436-437:"Oxidative stress may be the place where metabolic and oncogenic processes cross paths and may result in oncogenesis (102)." There is another reivew also discussed the role of oxidative stress in the development of tumors, especially breast cancer.(Please cite Chen et al. Semin Cancer Biol. 2020 Oct 6 doi:10.1016/j.semcancer.2020.09.012.)
2. The manuscript meed linguistic improvement.
Best
Author Response
Reviewer 2
Comments
In the manuscript by Shahid et al. the authors summarize recent data and research achievements on the relation and mechanisms between diabetes and breast cancer risk. I have just a few suggestions:
Some citations are missing:In page 11, line 436-437:"Oxidative stress may be the place where metabolic and oncogenic processes cross paths and may result in oncogenesis (102)." There is another reivew also discussed the role of oxidative stress in the development of tumors, especially breast cancer.(Please cite Chen et al. Semin Cancer Biol. 2020 Oct 6 doi:10.1016/j.semcancer.2020.09.012.)
- Thanks very much for the suggestion. We have added reference to work by Chen et al (Reference 147) in the revised paper.
The manuscript needs linguistic improvement.
- The revised manuscript has been thoroughly reviewed for spelling and grammatical errors (please see the track changes).

Reviewer 3 Report
Article is partially interesting, but unfortunately Authors tried describe many aspect of cancer and diabetes in one article, what is not good.
I suggest among others:
- remove from this article chapters 7. SPECIAL CONSIDERATIONS IN DIABETIC CANCER PATIENTS, and 8. SCREENING & DIAGNOSIS OF DIABETES IN CANCER PATIENTS. Especially chapter 8 is without details and I don't know what Authors wanted to present in it.
- In chapter 9. MANAGEMENT OF DIABETES IN CANCER PATIENTS, Authors describe management in e.g. cardiovascular complications or neuropathy, but it is not topic of this article. Authors should focus only on "diabetes in cancer patients", but not other diseases. Unfortunately, that's why the current descriptions are poor and without details. Hyperglycemia should be only one topic.
- In subchapter 9.1.5. Management of Infection, are very poor information. In human body are about 700-1000 bacteria, but Authors describe Staphylococcus aureus and Candida. In diabetic patients we observe many other, more important infections than S. aureus. Also, it is lack information about role of bacteria in cancer development (see and cite https://www.mdpi.com/2076-2607/7/1/20). What is role of Candida in sugar metabolism and carcinogenesis? This question with answers must be in this article. Why in diabetic patients, candidiasis is common, and oral cancer is often observed than in general population?
- In article lacks of chemical connection between diabetes and cancer, mechanisms of sugar metabolism in diabetic patients and impact on cancer nutrition and development (together with role of bacteria and fungi in this process). Please write about sugar-bacteria/fungi-cancer axis.
- In article lacks of figures: 1. with mechanisms of action diabetes-cancer, 2. with management of diabetes in cancer patients.
Author Response
Reviewer 3
Article is partially interesting, but unfortunately Authors tried describe many aspect of cancer and diabetes in one article, what is not good.
I suggest among others:
Remove from this article chapters 7. SPECIAL CONSIDERATIONS IN DIABETIC CANCER PATIENTS, and 8. SCREENING & DIAGNOSIS OF DIABETES IN CANCER PATIENTS. Especially chapter 8 is without details and I don't know what Authors wanted to present in it.
- Thanks very much for your suggestion. As the choice of anticancer treatment can be affected by underlying neuropathy or a cardiac or a renal disorder and is relevant to clinicians taken care of patients with cancer and diabetes we kept section 7 in the revised manuscript but have revised the title to “Factors that influence the choice of cancer therapy in patients with diabetes”. We have elaborated section 8 “Screening and Diagnosis of Diabetes” and have added additional information including definition of diabetes and relevance of post prandial blood glucose in patients treated with steroid.
In chapter 9. MANAGEMENT OF DIABETES IN CANCER PATIENTS, Authors describe management in e.g. cardiovascular complications or neuropathy, but it is not topic of this article. Authors should focus only on "diabetes in cancer patients", but not other diseases. Unfortunately, that's why the current descriptions are poor and without details. Hyperglycemia should be only one topic.
- We appreciate very much the comments. However, as mentioned above these comorbidities are major complications of diabetes and cancer therapy and are relevant to management of patients who are on active treatment or on surveillance during the cancer survivorship. Due to the scope of the review we have only provided key points. In the revised manuscript we have provided additional references to more extensive reviews in the management of cardiovascular and infectious complications in patients with diabetes.
In subchapter 9.1.5. Management of Infection, are very poor information. In human body are about 700-1000 bacteria, but Authors describe Staphylococcus aureus and Candida. In diabetic patients we observe many other, more important infections than S. aureus. Also, it is lack information about role of bacteria in cancer development (see and cite https://www.mdpi.com/2076-2607/7/1/20). What is role of Candida in sugar metabolism and carcinogenesis? This question with answers must be in this article. Why in diabetic patients, candidiasis is common, and oral cancer is often observed than in general population?
- We appreciate very much the comment and have elaborated management of infectious complications in the revised manuscript. We have added a new table (table 4) reviewing infection of main body sites and relevant pathogens in diabetic patients.
- We have reviewed possible etiology of high rate of candidiasis in diabetic patients and its association with the risk of cancer including oral cancer in the new section “Role of the Microbiome in Cancer and Diabetes”.
In article lacks of chemical connection between diabetes and cancer, mechanisms of sugar metabolism in diabetic patients and impact on cancer nutrition and development (together with role of bacteria and fungi in this process). Please write about sugar-bacteria/fungi-cancer axis.
- Thanks very much for this comments. We have added a new section “Role of the Microbiome in Cancer and Diabetes” in the revised manuscript under the mechanism of cancer risk in diabetic patients and have reviewed the role of microbiome in causing cancer.
In article lacks of figures: 1. with mechanisms of action diabetes-cancer, 2. with management of diabetes in cancer patients.
- Thanks very much for the suggestion. We have added two figures in the revised manuscript. Figure 1 shows the relationship between diabetes and cancer and figure 2 reviews the management of targeted treatment-related diabetes.

Reviewer 4 Report
The Authors provide an extensive overview on the relationship between diabetes and cancer and various aspects of the management of diabetes in cancer patients. It is a well-written review on a very relevant topic, even if it seems rather a repetition of other reviews published in recent years on the same subject.
Furthermore, there are some issues that the Authors should address.
Specific issues
- Paragraph 6 “Anti-cancer therapy & hyperglycemia”: some other cancer drugs with well-known diabetogenic potential should be mentioned too, such as diazoxide and L-Asparaginase, but also some common TKIs such as nilotinib and ponatinib. For further data on the adverse glycemic effects of cancer therapies and their management, please take into consideration papers such as: Gallo M et al, Metabolism 2018; Silvestris N et al, Crit Rev Oncol/Hematol 2020; Ariaans G. Cancer Treat Rev 2015.
- Paragraph 8 “Screening and diagnosis of DM in cancer patients”: many cancer patients with hyperglycemia are undiagnosed especially for not monitoring post-prandial glucose levels, such as people treated with corticosteroids (which may induce post-prandial hyperglycemia with “normal” basal glucose levels). Screening for PPG levels is mandatory in this population and should be highlighted in the paragraph.
- Paragraph 9.1.1 “Management of hyperglycemia”: Main recommendations on this topic do not agree with the Authors’ statement: “Patients may require a sliding insulin scale during cancer therapy”. Indeed, sliding scale insulin regimens should be discouraged in the inpatient setting and for cancer patients (see ADA, Diabetes Care in the Hospital: Standards of Medical Care in Diabetes-2021. Diabetes Care. 2021; Jacob P et al. QJM 2015).
- Paragraph 9.1.1 “Management of hyperglycemia”: The statement “If fasting glucose level is higher than 14 mm/l (250 mg/dl), cancer treatment interruption should be considered until blood glucose is better controlled” is also questionable: a good partnership and integrated care pathways between oncologists and diabetologists strongly reduce the need for cancer therapy interruption.
- Table 2: Glucosidase inhibitors and thiazolidinediones (rosiglitazone, pioglitazone) belong to two separate classes of diabetes drugs: please, split the line and modify accordingly (glucosidase inhibitors do not improve insulin sensitivity…)
- Table 2: GLP1-RAs, DPP-4 inhibitors and SGLT-2 inhibitors lines are incomplete (eg, lixisenatide, dulaglutide, semaglutide, alogliptin, ertugliflozin are not mentioned)
- Table 3, Line 4: the recommendation “Treatment with metformin, consider a second antihyperglycemic (e.g., sulfonylurea, etc)” for Grade 4 hyperglycemia (ie, >500 mg/dL) is not acceptable at all. Insulin therapy (intravenous or s.c.) is mandatory for this situation.
- Paragraph 10 “Future directions”: The statement “certain antidiabetic drugs may increase the risk of oncogenesis while others may decrease the risk” contrasts a bit with a previous one: “there is no definite evidence that they [anti-diabetes medications] either increase or decrease the risk of cancer” (see Paragraph 4.1).
- Paragraph 11 “Conclusion”: the topic of diabetic cancer survivors is very important, but it is only mentioned in the conclusions of the manuscript, and not faced earlier.
Minor issues
- Paragraph 4 “Link between DM and cancer”: I think that some other relevant papers on this topic deserve to be mentioned, such as: Sciacca L et al., Nutr Metab Cardiovasc Dis 2013; Mendonça FM et al., Metabolism 2015.
- Type of diabetes are somewhere indicated as “type 1” and “type 2” (correct), and somewhere as “type I” and “type II” (wrong). Please, unify.
- Many patients with cancer develop a secondary type of cancer (drug-induced, surgical-induced, type 3cDM, ), rather than type 2 DM: this aspect should be highlighted in the manuscript.
- Little typos should be corrected, eg: Page 4, Table 1, last line: “Bbinds”; Page 8, Line 363: “SGL-2”; Page 12, Line 459: “the use (of) certain drugs”. Please, check thoroughly the manuscript.
Author Response
Reviewer 4
Major comments
Paragraph 9.1.1 “Management of hyperglycemia”: Main recommendations on this topic do not agree with the Authors’ statement: “Patients may require a sliding insulin scale during cancer therapy”. Indeed, sliding scale insulin regimens should be discouraged in the inpatient setting and for cancer patients (see ADA, Diabetes Care in the Hospital: Standards of Medical Care in Diabetes-2021. Diabetes Care. 2021; Jacob P et al. QJM 2015).
- Thanks very much for this comment. We agree that sliding scale is a reactive management of hyperglycemia and not an optimal way of management of hyperglycemia. We have clarified it in the revised manuscript in the section of management of hyperglycemia.
Paragraph 9.1.1 “Management of hyperglycemia”: The statement “If fasting glucose level is higher than 14 mm/l (250 mg/dl), cancer treatment interruption should be considered until blood glucose is better controlled” is also questionable: a good partnership and integrated care pathways between oncologists and diabetologists strongly reduce the need for cancer therapy interruption.
- Thanks for this comments. This recommendations has been provided based on the general principle of withholding cancer treatment for treatment-related grade 3 toxicities till symptoms improve. However, we agree with the reviewer about the importance of an integrated care team and have rephrased the recommendation from “If fasting glucose level is higher than 14 mm/l (250 mg/dl), cancer treatment interruption should be considered until blood glucose is better controlled” to “If fasting glucose level is higher than 14 mm/l (250 mg/dl), cancer treatment interruption can be considered until blood glucose is better controlled”.
Table 2: Glucosidase inhibitors and thiazolidinediones (rosiglitazone, pioglitazone) belong to two separate classes of diabetes drugs: please, split the line and modify accordingly (glucosidase inhibitors do not improve insulin sensitivity…)
- Thanks very much for the correction. We have modified the table in the revised manuscript.
Table 2: GLP1-RAs, DPP-4 inhibitors and SGLT-2 inhibitors lines are incomplete (eg, lixisenatide, dulaglutide, semaglutide, alogliptin, ertugliflozin are not mentioned)
- Thanks for the suggestion. We have expanded the list of medication in the revised manuscript.
Table 3, Line 4: the recommendation “Treatment with metformin, consider a second antihyperglycemic (e.g., sulfonylurea, etc)” for Grade 4 hyperglycemia (ie, >500 mg/dL) is not acceptable at all. Insulin therapy (intravenous or s.c.) is mandatory for this situation.
- Thanks for the comments. We have clarified it in the revised table 3 and figure 2 about insulin administration for rapid correction of hyperglycemia with the subsequent use of metformin and other agents.
Paragraph 10 “Future directions”: The statement “certain antidiabetic drugs may increase the risk of oncogenesis while others may decrease the risk” contrasts a bit with a previous one: “there is no definite evidence that they [anti-diabetes medications] either increase or decrease the risk of cancer” (see Paragraph 4.1).
- Thanks very much for pointing it out. We have specified it that at the present time there is no definite causal relationship between antidiabetic drugs and risk of cancer.
Paragraph 11 “Conclusion”: the topic of diabetic cancer survivors is very important, but it is only mentioned in the conclusions of the manuscript, and not faced earlier.
- Thanks for the suggestion. We have added a brief section on survivorship care in diabetic patients and have also briefly highlighted it in the management of comorbid illness (Cardiovascular complication and hypertension) section.
Minor issues
Paragraph 4 “Link between DM and cancer”: I think that some other relevant papers on this topic deserve to be mentioned, such as: Sciacca L et al., Nutr Metab Cardiovasc Dis 2013; Mendonça FM et al., Metabolism 2015.
- Thanks for the suggestions, we have added the citations (Reference 27 and 33) in the revised manuscript.
Type of diabetes are somewhere indicated as “type 1” and “type 2” (correct), and somewhere as “type I” and “type II” (wrong). Please, unify.
- We have made the correction in the revised manuscript and have used Arabic numerical for consistency.
Many patients with cancer develop a secondary type of cancer (drug-induced, surgical-induced, type 3cDM, ), rather than type 2 DM: this aspect should be highlighted in the manuscript.
- Thanks very much for the suggestion. We have highlighted it in the revised manuscript.
Little typos should be corrected, eg: Page 4, Table 1, last line: “Bbinds”; Page 8, Line 363: “SGL-2”; Page 12, Line 459: “the use (of) certain drugs”. Please, check thoroughly the manuscript.
- Thanks for pointing it out. We have made the correction in the revised manuscript.

Round 2
Reviewer 2 Report
Authors made correction according to my previous suggestions. Strongly recommend for publishing.
Sincerely,